# Pharmacogenomic Testing: Clinical Evidence and Implementation Challenges

**DOI:** 10.3390/jpm9030040

**Published:** 2019-08-07

**Authors:** Catriona Hippman, Corey Nislow

**Affiliations:** 1Department of Psychiatry, Faculty of Medicine, University of British Columbia, Vancouver, BC V6T 2A1, Canada; 2BC Mental Health and Addictions Research Institute, 3rd Floor-938 West 28th Avenue, Vancouver, BC V5Z 4H4, Canada; 3Faculty of Pharmaceutical Sciences, University of British Columbia, 6619-2405 Wesbrook Mall, Vancouver, BC V6T 1Z3, Canada

**Keywords:** pharmacogenomics, clinical guidelines, implementation, drugs, medications, review

## Abstract

Pharmacogenomics can enhance patient care by enabling treatments tailored to genetic make-up and lowering risk of serious adverse events. As of June 2019, there are 132 pharmacogenomic dosing guidelines for 99 drugs and pharmacogenomic information is included in 309 medication labels. Recently, the technology for identifying individual-specific genetic variants (genotyping) has become more accessible. Next generation sequencing (NGS) is a cost-effective option for genotyping patients at many pharmacogenomic loci simultaneously, and guidelines for implementation of these data are available from organizations such as the Clinical Pharmacogenetics Implementation Consortium (CPIC) and the Dutch Pharmacogenetics Working Group (DPWG). NGS and related technologies are increasing knowledge in the research sphere, yet rates of genomic literacy remain low, resulting in a widening gap in knowledge translation to the patient. Multidisciplinary teams—including physicians, nurses, genetic counsellors, and pharmacists—will need to combine their expertise to deliver optimal pharmacogenomically-informed care.

## 1. Introduction

Pharmacogenomics is the study of how genes influence individuals’ responses to pharmacological treatments. Pharmacogenomics has broad applicability across clinical specialties and aspects of human health, as shown in Table 1. Despite the increased interest in genetics in the public sphere, driven by multiple factors, including direct-to-consumer testing, advances in genetic engineering, accumulating evidence of the importance of pharmacogenomics to successful pharmacological treatment, and the explosion of popular science journalism, the rate of adoption of pharmacogenomic testing in the clinical setting has been uneven. There is a significant gap in genomic literacy among medical doctors and other health care professionals. Indeed, only 1 in 10 physicians (*N* > 10,000, response rate: 3%) responding to a USA-based survey reported feeling confident in their knowledge of pharmacogenomics and its clinical application, less than 1/3 had ever ordered a pharmacogenetic test, and only 1/8 had recommended or ordered a test in the previous six months [1].

Overall, barriers to clinical implementation of pharmacogenomic testing fall into two broad categories: (1) answering the question of whether the testing should be performed at all, a point related to sufficiency of available evidence and cost-effectiveness, and (2) challenges associated with integration into the clinical system and work flow (such as the difficulty faced by clinical labs to comply with regulatory frameworks originally designed for non-genetic or single-gene tests) (Box 1). This review highlights some of the barriers to incorporating pharmacogenomic testing into clinical practice and considers how these barriers could be surmounted. This is not intended to be an exhaustive review; many barriers to clinical implementation of pharmacogenomic testing have already been covered well in the literature (see Box 1 for references).

Box 1Barriers to clinical implementation of pharmacogenetic testing.
**Barriers to clinical implementation (references in square brackets)**
1. Should testing be performed?
**Lack of evidence of clinical validity/utility of pharmacogenomic testing, including a lack of validated, pharmacogenomic-guided, treatment algorithms** [34,35,36,37,38,39,40,41]Lack of evidence demonstrating cost-effectiveness of pharmacogenomic testing (and consequent impact on cost to a public healthcare system, private health insurance companies, and out-of-pocket patient costs) [34,37,38,40,41,42,43,44,45]Lack of expertise amongst prescribing clinicians to determine whether a pharmacogenomic test is appropriate [35,37,42,46]Lack of recommendations from professional organizations, or changes to health policy, to support clinicians in determining whether a pharmacogenomic test is appropriate [37,40,45]Discrepancies between pharmacogenomic guidelines of different organizations [37,47]Perceived or actual financial conflicts of interest for authors of research/guidelines supporting the utility of pharmacogenomic testing**Ambiguity in how to clinically apply pharmacogenomic biomarker information in drug labels** [37,41,48]Lack of physician acceptance of pharmacogenomic testing [35,37,39,41,44,45,49]Lack of patient acceptance of pharmacogenomic testing (concerns regarding privacy, genetic discrimination, cost, etc.) [41,44,50,51,52]
2. Challenges to integration
Logistics of, and regulatory requirements for, performing pharmacogenomic testing in the clinical setting, including selection of genomic testing platform [34,35,38,44]Logistics of integrating pharmacogenomic test results into the electronic health record [34,35,37,38,44,45]Logistics of incorporating pharmacogenomic testing into the clinical workflow [35,41,42,43,44,49]Lack of standardized report formats for pharmacogenomic test results and inconsistency in practices for data storage and retrieval [35,37,38,42,53]Logistical and ethical issues regarding stewardship of pharmacogenomic test results, including responsibility for re-analyzing results in light of new evidence [37,42,43,44]**Complexity in pharmacogenomic test results, with attendant difficulties in interpretation—including the complex architecture of pharmacogenes such as *CYP2D6*, and the potential for interactions with other prescribed medications known to impact enzyme function** [34,37,41,45,49]Ambiguity in pharmacogenomic test results, such as variants of uncertain significance, particularly for non-White populations (for which there is a paucity of reference data), **and the lack of evidence regarding how to combine results from multiple pharmacogenes** [34,37,39,54]Lack of expertise amongst prescribing clinicians to interpret and manage a pharmacogenomic test result [34,35,37,38,42,43,46]Lack of support for clinicians to interpret and manage pharmacogenomic test results, such as inadequate information in drug labelling, pharmacogenomic practice guidelines, or decision support infrastructure [34,35,37,40,45]Discrepancies between pharmacogenomic guidelines of different organizations [37,48]Perceived or actual financial conflicts of interest for authors of research/guidelines supporting the utility of pharmacogenomic testingLack of physician acceptance of pharmacogenomic recommendations [35,37,49]Lack of ongoing patient engagement with pharmacogenomic testing (for communication of results) [44]**Identification of increased disease risks incidental to pharmacogenomic testing (e.g., *BRCA1/BRCA2* germline genetic test results to guide treatment also confer increased disease risk)** [35,37]
Note: Barriers in bold are those particularly relevant to pharmacogenomic testing (rather than genomic testing more generally).

## 2. Barrier 1: Should a Pharmacogenomic Test Be Ordered?

Not only is there increasing evidence in support of the clinical utility of pharmacogenetic testing, there is mounting evidence of the cost effectiveness of pharmacogenetic testing, albeit sometimes restricted to certain gene–drug combinations and/or specific populations [55,56,57,58]. However, the individual clinician attempting to answer this question for their patients is faced with a somewhat bewildering task. Keeping up to date with the tidal wave of pharmacogenomic evidence represents a considerable initial hurdle. As of June 2019, there were 309 drugs for which pharmacogenomics information is included in the labels—approved by one, or all, of the following agencies: the US Food and Drug Administration [59], the European Medicines Agency, Pharmaceuticals and Medical Devices Agency, Japan, and Health Canada/Santé Canada (according to the Pharmacogenomics Knowledgebase (PharmGKB) website (www.pharmgkb.org)). Unfortunately, the “pharmacogenomic biomarker” information provided in the labeling is highly variable in terms of detail provided, and can be ambiguous in terms of clinical guidance. For example, the label for the drug Iloperidone states that the “dose should be reduced by one-half for poor metabolizers of *CYP2D6*”, but it doesn’t explicitly state that testing for *CYP2D6* status should be performed prior to initiating therapy, or provide any support for how to order the testing or interpret the results.

It is perhaps unsurprising, then, that a significant discrepancy between drug labelling information and ordering practices has been observed. In a retrospective cohort study of prescription orders over three years (2011–2013) in the University of Washington Medicine health system, there were over 250,000 orders for drugs that had information in their labels regarding the drug’s association with germline pharmacogenomic variants [60]. Within those 268,262 orders, 8718 were for drugs whose label contained information for pharmacogenomic testing categorized as “recommended” or “required” by the specialist team who curate the PharmGKB website (as appears in the “Drug Label Annotations” section of the website). Of these 8718 medication orders, only 129 (1.5%) were associated with a pharmacogenomic test. This very low use of high-confidence pharmacogenomic information represents a lost opportunity to improve medication practices.

Compared to the information available in drug labelling, pharmacogenomic guidelines—published by organizations such as the Clinical Pharmacogenetics Implementation Consortium (CPIC) and the Dutch Pharmacogenetics Working Group (DPWG)—offer much clearer direction for clinicians. The CPIC and DPWG are two of the foremost organizations whose members have pharmacogenomic expertise and work to evaluate levels of evidence and release recommendations regarding the implementation of pharmacogenetic testing. A problem with relying on guidelines, however, is the limited number that are currently available. For the majority of gene-drug combinations, current levels of evidence are insufficient to support recommendations for changing clinical practice. Further, even when guidelines are available, clinical implementation of these recommendations is often not straightforward.

One such challenge arises when the guidelines from different organizations offer recommendations that are inconsistent. For example, CPIC and DPWG guidelines diverge in the magnitude of their recommendations in the case of the drug nortriptyline [4,32]. For individuals identified as *CYP2D6* intermediate metabolizers, CPIC guidelines recommend reduction of starting dose by 25%, while DPWG guidelines recommend reduction of starting dose by 40%. For individuals identified as *CYP2D6* poor metabolizers, CPIC guidelines recommend reduction of starting dose by 50%, while DPWG guidelines recommend reduction of starting dose by 60%. Although these differences in starting dose may not translate into large clinical differences, discrepancies such as this can cause confusion for the clinician and, of equal importance, do not inspire confidence in patients or providers. While details regarding exactly how the DPWG used pharmacokinetic data to develop their dosing recommendations are available in supplemental materials, the CPIC guidelines cite the DPWG guideline, but do not provide any further details regarding how they developed their recommendations for starting doses beyond citing their expert consensus process. The CPIC expert consensus process is described in detail in the literature [61]. However, an explication of the rationale for the choice of starting doses recommended, and the reason for the divergence from the cited DPWG guideline, are absent. A different problem with implementing guidelines—relevant to the issue of cost effectiveness—is when authors of the recommendations also appear to be seeking to gain financially from their research and practice guidelines through the pursuit of patents for the testing they recommend. This issue highlights the importance of the conflict of interest section of published literature in providing essential context for the interpretation and critical evaluation of practice guidelines. Both discrepancies between the guidelines of different pharmacogenomic organizations and perceived or real conflicts of interest for authors of pharmacogenomic guidelines undermine the credibility of these practice guidelines, which could contribute to the reluctance of prescribing clinicians to order pharmacogenomic testing.

## 3. Overcoming Barrier 1

Determining whether a given pharmacogenomic test should be ordered occurs on both an individual provider level and on a community level. At the individual level, work is underway across disciplines to raise awareness and educate clinicians about pharmacogenomics [34,53,62,63,64,65,66]. An excellent resource to help the busy clinician access the latest pharmacogenomic clinical guidelines and supporting evidence is the PharmGKB website [67]. The PharmGKB specialist team summarizes, interprets, and categorizes drug label information in terms of level of actionability of label information into “informative”, “actionable”, “recommended”, and “required”. The website also presents clinical annotations of the evidence and classifications of evidence related to gene–drug combinations using levels: 1a and 1b (high), 2a and 2b (moderate), 3 (low), and 4 (preliminary) (https://www.pharmgkb.org/page/clinAnnLevels). While these levels of evidence can provide useful context in terms of statistical significance, it is important to recognize that not all statistical associations are clinically actionable. The most important parameter for the individual clinician’s attention is evidence level 1a. This is the only level for which there is some evidence of clinical significance. At level 1a, gene–drug combinations are associated with either a clinical practice guideline and/or a known clinical implementation of testing. The PharmGKB website facilitates access to pharmacogenomic clinical practice guidelines by compiling them and making them freely available on the website, with curated highlights. It is important to note some limitations of the PharmGKB resource, however. For example, it is not immediately clear on the PharmGKB website whether a given “Very Important Pharmacogene” (VIP) is relevant for testing using saliva or whole blood samples, or only appropriate for testing tumor tissue. In other words, whether the known VIP variants are relevant to only somatic samples (that are thought to be uninvolved in a cancer), only cancer tissue samples, or both. Additionally, PharmGKB contains a mixture of both peer-reviewed and non-peer-reviewed content, consistent with its role as a clearing house for pharmacogenomic information. Accordingly, any gene–drug interaction information derived from this resource should be independently verified wherever possible. There are also limitations to the information available on the PharmGKB website with respect to supporting the interpretation of pharmacogenetic test results—discussed further under “Barrier 2”. The PharmGKB website is thus most useful to individual clinicians as an initial screening tool to identify whether pharmacogenomic testing may be worth considering for a given drug—by checking to see if there is a pharmacogenomic guideline with clinically actionable recommendations for that drug (https://www.pharmgkb.org/guidelineAnnotations). This page of the PharmGKB website compiles clinical practice guidelines from the CPIC, DPWG (English translation), Canadian Pharmacogenomics Network for Drug Safety (CPNDS), and other organizations, making it easy for clinicians to access guidelines for those gene–drug pairs that have been judged to be clinically significant by one of these organizations. Our strongest recommendation on an individual level is for a physician to consult with a pharmacogenomics specialist before ordering a pharmacogenomic test. Specialist healthcare providers are available for consultation—both in person and increasingly, online. Both pharmacists and genetic counsellors are trained to support fellow healthcare providers and patients to evaluate whether a given pharmacogenomic test is likely to yield meaningful and valuable information in the context of a given individual’s circumstances. The unique expertise of members of both professions is complementary to the skills and roles of other clinicians, such as physicians [68]. Given the complexity involved in implementing pharmacogenetic testing, and the diversity in skill sets and scopes of practice of healthcare professionals, multidisciplinary teams are ideal for providing optimal patient care [69,70,71].

On the community level, there are multiple national and international organizations dedicated to synthesizing emerging pharmacogenomic evidence and translating it into clinical practice guidelines. While there are currently a limited number of pharmacogenomic guidelines, the number of gene–drug combinations for which there are recommendations for clinical care—either in a guideline or a drug label—is steadily increasing (Table 2). There are currently (as of June 2019) 132 pharmacogenetic dosing guidelines for 99 drugs. These guidelines provide concrete, actionable recommendations for how to incorporate genetic test results into prescribing practices for gene–drug combinations with sufficient evidence, and they often include clinical decision protocols with descriptive flow charts and well-documented algorithms. While the majority of these guidelines focus on how to use pharmacogenomic information if it is already available (i.e., the patient and/or clinician are aware of the individual’s relevant genotype at the time of considering medication prescription), there is an increasing emphasis on creating guidelines for when to offer pharmacogenomic testing pre-emptively [72]. This latter trend is encouraging—it will expand the number of individuals that can benefit from pharmacogenomic information, but also increases the need for improved guidelines. There is movement at the pharmacogenomics expert community level to address the issue regarding inconsistency between the recommendations in the guidelines of different organizations. The organizations CPIC and DPWG in particular are committed to overcoming this barrier; they are working towards a goal to resolve discrepancies between their guidelines in order to facilitate implementation on a global scale [73,74]. With respect to the matter of conflict of interest in pharmacogenomic testing guidelines, the organization CPIC has imposed a very high threshold for accountability. Specifically, to promote transparency and accountability in their guideline process, CPIC requires contributors to its guidelines to disclose all possible conflicts of interest, which are reviewed by the CPIC Steering Committee as outlined in the Authorship Guidelines available on the CPIC website.

## 4. Barrier 2: Challenges with Implementing Pharmacogenetic Testing

Efforts to implement pharmacogenomic testing clinically encounter barriers that are shared with implementing other types of genomic testing as well as those which are specific to pharmacogenomics. We will first address those barriers that are common to clinical genomic implementation generally, and then those specific to pharmacogenomics, namely: the complex architecture of the major pharmacogene *CYP2D6*, the dearth of evidence available to combine results from multiple pharmacogenes when attempting to predict phenotype, and testing for pharmacogenes that are known to also confer disease risk.

### 4.1. Challenges Common to Implementing Genomic Testing Generally

There are many accepted options for clinically ascertaining genomic alleles and variants, including real-time polymerase chain reaction (RT-PCR), restriction length fragment polymorphism (RFLP) analysis, microarray, PCR followed by Sanger sequencing, and genome, exome or gene panel library preparation followed by Next Generation Sequencing (NGS)—also known as “massively-parallel sequencing”. A high-level summary comparing the advantages and disadvantages of these options, with particular attention to genomic testing of complex genes is provided in Table 3. Understanding the strengths and weaknesses of available genomic testing techniques is important to implementation considerations—especially with respect to the analysis of complex genes. The relative merits of different genomic testing platforms regarding complex genes, such as *CYP2D6*, is worthy of special attention in the context of pharmacogenomic implementation because *CYP2D6* is currently the pharmacogene that is paired with the most drugs for which pharmacogenomic guidelines recommend changes to medical management.

Given that each variant detection platform has non-overlapping strengths and weaknesses, combining different platforms enables the strengths of some to compensate for the weaknesses of others, and to provide some degree of orthogonal validation of the results. NGS options have been gaining in popularity as the cost of sequencing has fallen, and our knowledge of the human genome has expanded. NGS approaches can be highly cost effective because they enable testing of large numbers of individuals at many genetic loci simultaneously [75]. For detail regarding NGS approaches, see Box 2.

Box 2Best practices for clinical NGS.An NGS test minimally involves: (a) specimen collection and storage, (b) nucleic acid extraction, (c) NGS library preparation, (d) sequencing and base calling, (e) sequence alignment/mapping, (f) variant calling and annotation, (g) variant evaluation and (h) report generation. After preparation of DNA for analysis (the in vitro steps necessary to transform the blood/saliva/tissue sample into pure DNA, followed by amplification of the sample—(b) and (c)), further in silico steps are necessary to interpret the resulting data ((d), (e), (f), and (g)), including alignment—the use of short-read sequencing mapping tools (used in combination with a standard —well characterized-genome, for reference, and a quality control process) and then evaluation of the aligned sequences to make meaning of identified variants for use in the clinical context (variant interpretation). Variant interpretation uses information from the published literature, databases documenting the clinical significance of variants (e.g., ClinVar [86]), tools known as functional prediction programs (particularly when variants are not documented in the literature or databases), and clinical information obtained from the patient and family.Technical validity must be established to assure the reliability and robustness of results prior to clinical implementation of an NGS assay. Performance specifications that are required to demonstrate technical validity of an NGS assay include: Accuracy, precision, limit of detection (LoD), and analytical specificity. Accuracy is the degree of concordance between a sequence obtained from the test and the same sequence determined by a comparator method or a reference. Precision is also known as reproducibility, and refers to the impact of, e.g., different operators, operating conditions, days of measurement, instruments, etc., on the same sample. The limit of detection refers to the upper and lower bounds of input DNA sample quantity that were determined during the validation process to produce accurate results 95% of the times that the assay was executed. Specificity refers to the assay’s capacity to discriminate and identify only each variant present, avoiding false positives and false negatives that may arise through cross-reactivity and contamination.A fundamental step in evaluating the accuracy of an NGS assay as part of assay validation is testing samples that have been well characterized and can thus provide a gold standard result for comparison and determination of assay performance (sensitivity, specificity, etc.). Biological reference samples that can be used for this purpose are available through the nonprofit Coriell Institute for Medical Research or the Genome in a Bottle Consortium [87]. In addition to the use of biological reference materials for assay validation, an often-overlooked NGS technique for the validation of results uses so-called “spike-in controls” [88]. Spike-in controls are usually non-human or synthetic samples of DNA or RNA, which have known properties such as length, sequence, and structure. Synthetic spike-in controls can be designed with particular features for a given genetic test and NGS platform, making this approach very flexible. Another advantage of using spike-in controls is that they can be mixed directly with patient samples for the NGS library preparation stage prior to sequencing, unlike biological reference samples, which need to be separate for this stage to avoid contamination of the patient samples and attendant difficulties in interpretation. Spike-in controls thus provide a quality assurance mechanism to evaluate sources of random and systematic error in the whole NGS pipeline. The drawback with spike-in controls is the greater difficulty (compared to biological reference samples) in demonstrating “commutability”—i.e., the ability to perform comparably to patient samples—before use.In practice, once an assay has been validated for clinical use, parameters cannot be changed without re-validation. This also represents a major challenge when NGS technology is continuously evolving, which relates to yet another challenge, namely that reproducing results may not be possible over time given the rapidity with which the technology is changing. Further, storage of results is difficult, and it remains unclear exactly what data should be stored. In the first decade of NGS, for example, raw images were saved. However, given the 100–1000-fold increase in data collected/assayed, the size of the raw image result files is prohibitive for long-term storage in the clinical setting. Even without images, .bcl files, for example, can be cumbersome to manipulate, and the significantly smaller .fasta file format has become the de facto standard.Best practice guidelines for clinical NGS have been published by the USA CDC working group Nex-StoCT II [89], CAP/AMP [90,91], ACMG [92], FDA [93], the UK organization ACGS [94], the Dutch Genome Diagnostic Laboratories [95], and the Korean Society of Pathologists [96]. While some older guidelines recommend orthogonal validation of all reported variants, newer guidelines do not include this requirement, likely due to mounting evidence that this is unnecessary for most variants; that is, labs can now be more selective about which variants they choose to confirm with an orthogonal method [97,98,99]. Guidelines from countries that have universal healthcare (UK, the Netherlands, Korea) recommend data sharing, e.g., submission of variant data to open source platforms such as ClinVar. One organization in the USA—ACMG—also includes this recommendation in their guidelines.Gargis et al. (2016) [100] provide an overview of strategies for operationalizing NGS assay validation for the clinical setting. For definitions of key terms relevant to NGS (such as “base call quality score”, “read depth”—a.k.a. “coverage”, “variant read number”, “variant allele frequency”, “variant quality scores”, and “strand bias”), written in language accessible to non-expert clinicians, see Strom [101]. For a visual representation of the process of NGS, see Figure 2, p. 467, in Moorcraft et al. [85], or Figure 1, p. 125, in Oliver et al. [102].

There are significant challenges associated with routine clinical implementation of NGS technologies. Clinical laboratories do not typically have the infrastructure—e.g., computational servers and databases—necessary for processing NGS results, and personnel of clinical laboratories do not typically have the skill required for interpreting NGS results [100]. The validation process for laboratories that is required prior to clinical use of any test (to be in line with national regulations, e.g., Clinical Laboratory Improvement Amendments (CLIA) and the College of American Pathologists (CAP) in the USA, as well as the Standards Council of Canada) focuses on measures such as sensitivity, specificity, positive predictive value, and negative predictive value—which are relatively simple when there is one target analyte being measured, but are much less straight-forward to operationalize when characterizing large numbers of targets, for example, when using NGS. Given the heterogeneity in NGS-based clinical tests, with differences in reagents, labware, instruments and software, any two NGS-based tests will likely differ in design and workflow. These technical differences can have important effects on the resulting data in terms of parameters such as sensitivity and positive predictive value. According to a recent guidance document published by the USA’s FDA, the “FDA is unaware of any existing, comprehensive standards for analytical validation applicable to NGS-based tests intended to aid in the diagnosis of suspected germline diseases that it believes could be used to help provide a reasonable assurance of the safety and effectiveness of these tests” [93]. While the FDA has approved a few single-gene NGS-based tests, they have not-to date-classified more general NGS tests.

Beyond the technical challenges of analysis and interpretation, two issues in particular stand out as presenting additional ethical challenges, namely, the potential for identification of a variant of uncertain significance (VUS) and variants unrelated to the reason for the testing (a.k.a. incidental or secondary findings). These issues are less of a concern when using NGS for targeted gene panel testing—in this case, it is usual to restrict the assay space and the subsequent analysis to only variants of known clinical significance within targeted genes of interest. However, genome sequencing (GS) and exome sequencing (ES) are being used with increasing frequency—especially for diagnostic purposes amongst patients presenting with severe and rare phenotypes where targeted genetic testing has not identified the cause of the patient’s condition. While it is possible to restrict the analysis of GS/ES to only variants of known significance within targeted genes of interest, doing so is ethically and logistically contentious [103,104,105,106,107].

Current knowledge of the variants within the human genome and their functional consequences (or lack thereof) is far from perfect. This is especially true when considering non-White populations, which are vastly under-represented in the existing databases. Indeed, when sequencing is performed in the more exploratory fashion of research-based GS/ES (without pre-determined targets of interest), the identification of a VUS is exceedingly common (almost a guarantee). Ethical, legal, and logistical questions abound with respect to the reporting of a VUS, the management of clinical care in response to a reported VUS, and where to place the burden of responsibility for re-evaluation of variants in response to the accumulation of evidence over time (enabling the reclassification of a variant from a VUS to either benign or pathogenic—and sometimes the reclassification of variants previously reported to be benign or pathogenic). One recent example of how the science is well ahead of the regulatory infrastructure—a lawsuit adjudicated by the South Carolina Supreme Court [Williams vs. Quest/Athena]—also illustrates the complexity of the issues that relate to the reporting of variants of uncertain significance in clinical care [108]. This is a single example, but it is reasonable to expect that it represents an unanticipated consequence of the broader application of genomic—and pharmacogenomic—testing. Further, GS/ES will likely identify incidental variation, such as carrier status for recessive conditions. Members of professional genetics organizations continue to debate whether, and under what circumstances, laboratories and clinicians are justified or obligated to report (in the test results), and return to patients, these incidental findings [105,109].

### 4.2. Challenges for Pharmacogenomics Implementation: CYP2D6

The pharmacogene with the most drugs for which pharmacogenetic-guided therapy is recommended is *CYP2D6.* This gene also happens to have a very complex architecture, with over 100 variants described to date (www.pharmvar.org), including SNPs, CNVs, small insertions or deletions, and larger scale gene rearrangements (such as the formation of hybrid genes with the nearby pseudogene *CYP2D7*). This complexity poses challenges in the analysis and interpretation of results for *CYP2D6*. Additional considerations are, therefore, important when planning analysis of *CYP2D6* (Table 3). A thorough discussion of the challenges associated with *CYP2D6* testing is provided by *CYP2D6* expert, Gaedigk [110].

### 4.3. Challenges for Pharmacogenomics Implementation: Combining Results from Multiple Genes

As with other genetic testing, the value in pharmacogenetic results lie in their power to predict phenotype and outcomes, i.e., what to expect in terms of medication response. As is typical in genetic research, initial pharmacogenetic investigations have focused on one gene and one drug. A difficulty, however, in applying results from one gene in relation to one drug is that drugs are frequently not taken in isolation and that—usually—there is more than one gene in the drug response pathway. For some drugs, algorithms have been developed to incorporate results from multiple genes in guiding drug prescribing—most notably, results from *CYP2C9*, *VKORC1*, *CYP4F2*, and rs12777823 in guiding warfarin starting dose [9]. However, warfarin is one of only a very small minority of drugs for which guidelines are available that incorporate results from multiple genes (amitriptyline and other TCAs [32]; the anticonvulsants carbamazepine [21] and phenytoin [20]; and the antineoplastics daunorubicin and doxorubicin [28]). Furthermore, even in these well-characterized cases, the fact remains that individuals are often taking multiple prescriptions—with the potential for synergistic/antagonistic combinations, for which current algorithms do not account.

### 4.4. Challenges for Pharmacogenomics Implementation: When Pharmacogenes Are Also Disease Risk Genes

It is vital to acknowledge that some genes that have been identified as pharmacogenes, for which pharmacogenomic guidelines or drug labels recommend changes to medical management in the context of certain genetic variants, are also known to confer increased disease risk. Most notable amongst these are *BRCA1* and *BRCA2*. The drug labels of the USA FDA for olaparib and rucaparib require *BRCA* genetic testing (germline or somatic) prior to their use in the treatment of breast, ovarian, fallopian tube, or peritoneal cancer. Germline genetic testing for *BRCA1* and *BRCA2* has much broader implications beyond guiding treatment, however. As outlined in the practice guideline by the National Society of Genetic Counselors [111], germline pathogenic variants in *BRCA1* and *BRCA2* confer increased risk for cancers for individuals in whom variants have been identified, and also for their family members. We strongly recommend that specialists such as genetic counsellors be involved in any germline genetic testing for *BRCA1* and *BRCA2*, and for any other genes that are known to confer increased disease risk in addition to their pharmacogenomic role(s).

## 5. Overcoming Barrier 2

Regarding technical challenges common to all genomic testing implementation, best practice guidelines for clinical NGS—including for the identification of pharmacogenetic variants—have been published by the USA CDC working group Nex-StoCT II [89], CAP/AMP [90,91], ACMG [92], FDA [93], the UK organization ACGS [94], the Dutch Genome Diagnostic Laboratories [95], the Korean Society of Pathologists [96], amongst others. Consistent recommendations include:Rigorous sample tracking methodology,precise documentation of the NGS process used to generate results (e.g., assay platform, software version and settings, reference genome sequence ID—including version number, quality metrics—both methods used for, and results of, such quality evaluation),use of standardized, widely-accepted nomenclature (e.g., available from the Human Genome Variation Society (http://www.hgvs.org/mutnomen) and PharmVar (https://www.pharmvar.org/)) for variant identification, classification, and reporting,clear documentation of the limitations of the clinical NGS in the test report,storage of the variant (VCF) files at a minimum, and the alignment mapping (BAM/SAM) files if possible,validation of the NGS pipeline, and re-validation following any parameter changes (e.g., software updates; re-validation may be in whole or in part—depending on the anticipated types of errors associated with the change),ongoing quality assurance testing—such as proficiency testing,compliance with all relevant legal and policy frameworks (local, provincial, national), andinvolvement of highly qualified personnel with certification from relevant professional bodies.

The most recent guidelines, from the USA FDA, place an emphasis on the standards required for an NGS report. In summary, they describe the minimum features of a report as including “a prominently-placed list of pathogenic or actionable variants on the first page of a test report” [30] (p. 29). If variants of unknown significance are included, a statement that their clinical relevance is not known should be included, as should a roster of which classes of variants are not included. Finally, test limitations, including regions that failed sequencing as well as limitations to variant evaluation, should be included. These new frameworks represent an encouraging development that can help to address both interpretation and implementation challenges.

With respect to the ethical issues common to all genomic testing implementation, if GS/ES ordered by a clinician should identify pharmacogenomic variants with known clinical implications, the case for returning these results to patients is strong—there is the potential to both enhance treatment outcomes (according to the ethical principle of beneficence) and minimize risk for harm (according to the ethical principle of non-maleficence). Not only that, when clinicians are already ordering GS/ES, adding the interpretation of pharmacogenes arguably makes both clinical and economic sense. For clinicians facing ambiguous or unexpected results, pharmacists and genetic counsellors are, again, excellent resources to support their fellow healthcare providers in the interpretation and management of such results. Ideally, GS/ES should be performed (and analyzed) only in the context of a multidisciplinary clinical team that includes medical geneticists and/or genetic counsellors who have the expertise required to interpret and manage the full spectrum of genetic test results—including the identification of variants (primarily or secondarily) with potential impact on disease risk and/or drug metabolism.

With reference to the logistical challenges associated with implementing pharmacogenomic testing, for characterizing complex pharmacogenes like *CYP2D6*, Gaedigk brings attention to the limitations of using single methods and advocates bringing multiple methods to bear on a sample [110]. Gaedigk concurs with the emerging consensus and recommends the use of a *CYP2D6*-specific amplicon—generated using XL-PCR as a first step in the analysis—for subsequent genotyping, which enables the detection of deletions, duplications, and hybrid genes. Gaedigk also recommends that multiple regions along the *CYP2D6* gene be probed in order to evaluate copy number variation—quantifying the number of duplications, for example, and also enabling exploration of more complex gene rearrangements and hybrid genes. In terms of combining results from multiple pharmacogenes, particularly given the greater cost-effectiveness of multi-gene panels, there is an urgent need for further research elucidating the relative contributions of different genes along the metabolic pathways for different drugs.

## 6. Conclusions

The landscape of pharmacogenomic testing is rapidly evolving. While barriers to the implementation of pharmacogenomic testing into clinical practice are multifaceted—systemic, individual, legal, logistical, knowledge-based, values-based—there are resources available, such as PharmGKB, practice guidelines, pharmacists, and genetic counsellors, to support clinicians to implement this testing in their practice.

## 7. Future Perspective

Over half a century ago saw the publication of the first pharmacogenetics textbook (monograph) [112], and authors have been touting the promise of pharmacogenetics ever since. We, too, believe that pharmacogenetics holds considerable, largely untapped, potential to optimize medication selection, dosing, and to avoid adverse drug responses. It is difficult to predict how quickly and broadly pharmacogenomic testing will be adopted in different countries and contexts, but it is very promising that large scale implementation efforts to address health-systems integration challenges associated with pharmacogenomic testing are currently underway on a research basis in the USA (e.g., PGRN’s Translational Pharmacogenomics Program [34,53]; the Right Drug, Right Dose, Right Time protocol [113]; the PREDICT program [114]; and NIH’s All of Us project—https://allofus.nih.gov/about/scientific-opportunities), and Europe (www.upgx.eu) [115].

It is clear that

the actionable evidence for pharmacogenomics will continue to accumulate,the technology will continue to advance and become more accessible, andcosts will continue to drop.

We suggest that, in addition to continuing the excellent work of generating evidence, synthesizing evidence and putting it into context in practice guidelines, and ongoing efforts to increase accessibility of the evidence for clinicians, the pharmacogenomics community needs to do everything we can to identify and engage allies—across healthcare disciplines—in pushing for systemic and cultural change. It is our collective responsibility to ensure that the potential health benefits of pharmacogenomics reach beyond early-adopter and privileged niches. In striving towards this goal, it is important to note that the vast majority of available data have been collected from primarily White populations [116]. Given that we know that the reference genome and associated variants are missing over 300 million bases for other populations [117], it will be essential to continually update the statistics on population-specific variant frequencies, and reconsider implications for guidelines and clinical practice in light of these new data.

## 8. Executive Summary

Pharmacogenomics is relevant to every aspect of human health.Barriers to incorporating pharmacogenomic testing into clinical practice include: low genomic literacy amongst physicians; drug labelling information that is difficult to interpret and/or out-of-date; clinical guidelines for pharmacogenetic testing that are sometimes discrepant (between organizations), and occasionally may be biased; and technical as well as logistical challenges in pharmacogenetic analysis and interpretation of results.A growing number of guidelines (132 available on the PharmGKB website, as of June 2019) provide recommendations for the use of pharmacogenetic testing to guide clinical care.PharmGKB facilitates access to guidelines for clinicians.CPIC and DPWG are committed to resolving discrepancies between guidelines.Work is underway in multiples countries worldwide to address the logistical challenges of integrating pharmacogenetic testing into healthcare systems—with large-scale implementation studies underway in the USA and Europe, and best practice guidelines for clinical next generation sequencing published by numerous organizations.Working as part of a multidisciplinary team with pharmacists and genetic counsellors can help with managing the complex challenges of determining whether to order a pharmacogenomic test, and then interpreting and acting on the results. Find a genetic counsellor using directories available through the USA-based National Society of Genetic Counselors website (https://www.nsgc.org/page/find-a-genetic-counselor) or the Canadian-based Canadian Association of Genetic Counsellors website (https://www.cagc-accg.ca/?page=225).

## Figures and Tables

**Table 1 jpm-09-00040-t001:** Connections between clinical specialty areas, drugs, and VIPs for which guidelines for clinical implementation are available on the PharmGKB site (under “Clinical Guideline Annotations”). VIP = Very Important Pharmacogene (as defined by PharmGKB).

Clinical Specialty Area	Drug Class	Drug (s)	Relevant VIP (s)	Associated Guideline (s)
Anesthesiology	Anesthetic agents and muscle relaxants	Desflurane, enflurane, halothane, isoflurane, methoxyflurane, sevoflurane, succinylcholine	*CACNA1S,*	CPIC [2]
			*RYR1*	
Cardiology	Anti-arrhythmics	Flecainide, propafenone	*CYP2D6*	DPWG [3,4]
	Beta blockers	Metoprolol	*CYP2D6*	DPWG [3,4]
	Statins (lipid management)	Simvastatin	*SLCO1B1*	CPIC [5]
				DPWG [3]
Dermatology	Anti-fungal (Aspergillosis, Candidiasis)	Voriconazole	*CYP2C19*	CPIC [6]
				DPWG [3,4]
Endocrinology	Hormonal contraceptives (estrogen-containing)	Combined injectable contraceptive, contraceptive patch, NuvaRing, oral contraceptive pill	*F5*	DPWG [3,4]
	Protein “potentiator” (cystic fibrosis treatment)	Ivacaftor	*CFTR*	CPIC [7]
Gastroenterology	Anti-fungal (Candidiasis)	Voriconazole	*CYP2C19*	CPIC [6]
				DPWG [3,4]
	Anti-emetic	Ondansetron, tropisetron	*CYP2D6*	CPIC [8]
	Protein “potentiator” (cystic fibrosis treatment)	Ivacaftor	*CFTR*	CPIC [7]
	Proton pump inhibitors	Lansoprazole, omeprazole, pantoprazole	*CYP2C19*	DPWG [3,4]
Gynecology	Anti-fungal (Candidiasis)	Voriconazole	*CYP2C19*	CPIC [6]
				DPWG [3,4]
Hematology	Anti-thrombotic (anticoagulant/antiplatelet)	Acenocoumarol, clopidogrel, phenprocoumon, warfarin	*CYP2C19, CYP2C9, CYP4F2, VKORC1*	DPWG [3,4]
				CPIC [9,10]
				CPNDS [11]
Immunology	Anti-retroviral (HIV treatment)	Abacavir, atazanavir	*HLA-B, UGT1A1*	CPIC [12,13]
				DPWG [3,4]
	Anti-viral (hepatitis C, RSV, viral hemorrhagic fever treatment)	Peginterferon alfa-2a, peginterferon alfa-2b, ribavirin	*HLA-B, IFNL3*	CPIC [14]
				DPWG [4]
	Immunosuppressant (eczema, rheumatoid arthritis treatment, lowers risk of organ rejection following transplant)	Azathioprine, mercaptopurine, tacrolimus, thioguanine	*CYP3A5, TPMT*	CPIC [15,16]
				DPWG [3,4]
Nephrology	Anti-gout agent (also kidney stones treatment)	Allopurinol, rasburicase	*G6PD, HLA-B*	CPIC [17,18]
				American College of Rheumatology [19]
Neurology	Anti-convulsant	Carbamazepine, phenytoin, oxcarbazepine	*CYP2C9, HLA-A, HLA-B*	CPIC [20,21]
				CPNDS [22]
				DPWG [3,4]
	Anti-fungal (CNS fungal infections treatment)	Voriconazole	*CYP2C19*	CPIC [6]
				DPWG [3,4]
	Opioid analgesics	Codeine, tramadol	*CYP2D6*	CPIC [23]
				DPWG [3,4]
				CPNDS [24]
Oncology	Anti-neoplastics	Capecitabine, cisplatin, daunorubicin, doxorubicin, fluorouracil, irinotecan, tamoxifen, tegafur	*BRCA1, CYP2D6, DPYD, RARG,* *SLC28A3, TPMT, UGT1A6, UGT1A1*	CPIC [25,26]
				DPWG [3,4]
				CPNDS [27,28,29]
				French Group of Clinical Onco-pharmacology & National Pharmacogenetics Network [30]
Ophthalmology	Anti-fungal (Aspergillosis)	Voriconazole	*CYP2C19*	CPIC [6]
				DPWG [3,4]
Otolaryngology	Anti-fungal (Candidiasis)	Voriconazole	*CYP2C19*	CPIC [6]
				DPWG [3,4]
Psychiatry	Anti-convulsants	Carbamazepine, phenytoin, oxcarbazepine	*CYP2C9, HLA-A, HLA-B*	CPIC [20,21]
				CPNDS [22]
				DPWG [3,4]
	Anti-depressants	SNRI: venlafaxine; SSRI: citalopram, escitalopram, fluvoxamine, paroxetine, sertraline; TCA (tricyclic): amitriptyline, clomipramine, desipramine, doxepin, imipramine, nortriptyline, trimipramine	*CYP2C19, CYP2D6*	CPIC [31,32]
				DPWG [3,4]
	Anti-psychotics	Atypical: aripiprazole, Typical: haloperidol, zuclopenthixol	*CYP2D6*	DPWG [3,4]
	Impulse control (ADHD treatment)	SNRI: atomoxetine	*CYP2D6*	CPIC [33]
				DPWG [3,4]
Respirology	Anti-fungal (Aspergillosis)	Voriconazole	*CYP2C19*	CPIC [6]
				DPWG [3,4]
	Protein ‘potentiator’ (cystic fibrosis treatment)	Ivacaftor	*CFTR*	CPIC [7]
Rheumatology	Anti-gout agent (also treats kidney stones, high uric acid levels secondary to cancer treatment)	Allopurinol, rasburicase	*G6PD, HLA-B*	CPIC [17,18]
				American College of Rheumatology [19]
Urology	Anti-gout agent (also kidney stones treatment)	Allopurinol, rasburicase	*G6PD, HLA-B*	CPIC [17,18]
				American College of Rheumatology [19]

Note: Table organized for ease of navigation by clinical specialty, so drug classes, drugs, and VIPs may appear in multiple rows. Gene–drug pairs are not included if associated guidelines conclude with no actionable recommendations.

**Table 2 jpm-09-00040-t002:** List of PharmGKB VIPs (*n* = 24) for which guidelines or drug labels recommend change(s) to medical management on the basis of clinical genetic testing (single gene DNA) results (focused only on germline testing; not including somatic testing, mRNA testing, or cytogenetic testing for large-scale chromosomal structural variants).

VIP	Clinical Genetic Testing ^1,2^	Drug (Guideline/Drug Label Organizations ^3^)	Clinical Impact
*BRCA1*	Genetic testing for *BRCA1* mutations	Olaparib, rucaparib (FDA drug label)	Targeted treatment specific to genetic status
*BRCA2*	Genetic testing for *BRCA2* mutations	Olaparib, rucaparib (FDA drug label)	Targeted treatment specific to genetic status
*CACNA1S*	Genetic testing for *CACNA1S* mutations	Desflurane, enflurane, halothane, isoflurane, methoxyflurane, sevoflurane, succinylcholine (CPIC [2])	Alternate choice of medication to prevent serious ADR (risk of death)
*CFTR*	Genetic testing for presence of *CFTR* G551D, F508del variants (+32 other variants now approved—found on ivacaftor drug label)	Ivacaftor (CPIC [7]), lumacaftor (when in formulation with ivacaftor) (FDA drug label)	Targeted treatment specific to genetic status
*CYP2C19*	Genetic testing for presence of increased and decreased function alleles	Clopidogrel (DPWG [3,4], CPIC [10]) Amitriptyline, clomipramine, doxepin, imipramine, trimipramine (CPIC [32]—all tricyclic antidepressants listed, DPWG [3,4])—only imipramine) Citalopram, escitalopram, sertraline (CPIC [31], DPWG [3,4]) Voriconazole (CPIC [6], DPWG [3,4])	Dosing adjustment/alternate choice of medication (risk of poor efficacy/ADRs)
		Lansoprazole, omeprazole, pantoprazole (DPWG [3,4])	Increase attention/monitoring dose
*CYP2C9*	Genetic testing for presence of decreased function alleles	Phenytoin (CPIC [20], DPWG [3,4])	Dosing adjustment to prevent serious ADR
	Genetic testing for presence of decreased function alleles	Warfarin (CPIC [9], CPNDS [11], DPWG [3])	Dosing adjustment for optimal efficacy (avoiding excessive bleeding/clotting)
*CYP2D6*	Genetic testing for presence of increased and decreased function alleles (recommendation may be based on genotype activity score)	Amitriptyline, also likely applicable to other TCAs: Clomipramine, desipramine, doxepin, imipramine, nortriptyline, trimipramine (CPIC [32]—as listed, DPWG [3,4]—only amitriptyline, clomipramine, doxepin, imipramine, nortriptyline) Aripiprazole, haloperidol, pimozide, zuclopenthixol (DPWG [3,4]), fluvoxamine (CPIC), paroxetine (CPIC [31]—both SSRIs listed, DPWG [3,4]—only paroxetine) Venlafaxine (DPWG [3,4]) Codeine (CPIC [23], DPWG [3,4], CPNDS [24]), tramadol (DPWG [3,4]) Flecainide, propafenone (DPWG [3,4]) Metoprolol (DPWG [3,4]) Tamoxifen (CPIC [26], DPWG [3,4], CPNDS [29]) Eliglustat (DPWG [3]) Tetrabenazine (FDA drug label)	Dosing adjustment/alternate choice of medication (risk of poor efficacy/ADRs)
		Ondansetron, tropisetron (CPIC [8])	Alternate choice of medication to reduce risk of poor efficacy for UMs
		Atomoxetine (CPIC [33], DPWG [3,4])	Increase attention/monitoring dose
*CYP3A5*	Genetic testing for presence of “normal” function and decreased function alleles	Tacrolimus (CPIC [16], DPWG [3,4])	Dosing adjustment to reduce risk of poor efficacy
*CYP4F2*	Genetic testing for presence of *CYP4F2*3* allele	Warfarin (CPIC [9])	Dosing adjustment for optimal efficacy (avoiding excessive bleeding/clotting)
*DPYD*	Genetic testing for presence of decreased function alleles (recommendation based on genotype activity score)	Capecitabine, fluorouracil, tegafur (CPIC [25]—only capecitabine and fluorouracil, DPWG [3,4]—all three anti-neoplastics listed)	Dosing adjustment/alternate choice of medication (risk of ADR—death)
*DMD*	Genetic testing for presence of *DMD* mutation that is amenable to exon 51 skipping	Eteplirsen (FDA drug label)	Targeted treatment specific to genetic status
*F5*	Genetic testing for *F5* alleles	Estrogen-containing hormonal contraceptives (DPWG [3,4])	Alternate choice of contraceptive method to prevent serious ADR (venous thrombo-embolism)
*G6PD*	Genetic testing for presence of decreased function (class I, II, or III) alleles [x-linked—males 1 allele, females—2 alleles; if ambiguous result or female heterozygote—enzymatic testing to confirm activity levels]	Rasburicase (CPIC [18]) Pegloticase (FDA drug label, European Medicines Agency drug label) Primaquine (FDA drug label)	Alternate choice of medication to prevent serious ADR (acute hemolytic anemia)
*HLA-A*	Genetic testing for presence of *HLA-A*31:01* variant	Carbamazepine (CPIC [21], CPNDS [22])	Dosing adjustment to prevent serious ADR (SCAR)
*HLA-B*	Genetic testing for presence of *HLA-B*15:02* variant	Carbamazepine (CPIC [21], CPNDS [22]), phenytoin (CPIC [20]), oxcarbazepine (CPIC [21])	Dosing adjustment to prevent serious ADR (SCAR)
	Genetic testing for presence of *HLA-B*57:01* variant	Abacavir (CPIC [12], DPWG [3,4])	Dosing adjustment/alternate choice of medication (risk of poor efficacy/ADR—SCAR)
	Genetic testing for presence of *HLA-B*58:01* variant	Allopurinol (CPIC [17], American College of Rheumatology [19])	Dosing adjustment to prevent serious ADR (SCAR)
*IFNL3*	Genetic testing for presence of *IFNL3* (IL28B) variant (rs12979860)	Peginterferon alfa-2a, peginterferon alfa-2b, ribavirin (CPIC [14])	Anticipated efficacy—consider in context of SDM and likely side effects
*POLG*	Mitochondrial genetic testing for *POLG* mutations	Divalproex sodium (FDA drug label, Health Canada/Santé Canada drug label)	Alternate choice of medication to prevent serious ADR (acute liver failure and death)
*RARG*	Genetic testing for presence of *RARG* rs2229774 variant	Daunorubicin, doxorubicin (CPNDS [28])	Pediatric patients: Dosing adjustment to prevent serious ADR (cardiotoxicity)
*RYR1*	Genetic testing for *RYR1* mutations	Desflurane, enflurane, halothane, isoflurane, methoxyflurane, sevoflurane, succinylcholine (CPIC [2])	Alternate choice of medication to prevent serious ADR (risk of death)
*SLCO1B1*	Genetic testing for presence of C allele at *SLCO1B1* rs4149056	Simvastatin (CPIC [5], DPWG [3])	Dosing adjustment/alternate choice of medication to prevent serious ADR (myopathy)
*TPMT*	Genetic testing for presence of decreased function alleles	Azathioprine, mercaptopurine, thioguanine (CPIC [15], DPWG [3,4])	Dosing adjustment/alternate choice of medication (risk of poor efficacy/ADRs)
	Genetic testing for presence of *TPMT *2, *3A, *3B, *3C* alleles	Cisplatin (CPNDS [27])	Pediatric patients: Dosing adjustment to prevent serious ADR (ototoxicity)
*UGT1A1*	Genetic testing for presence of two decreased function alleles	Atazanavir (CPIC [13])	Dosing adjustment to prevent serious ADR (jaundice)
	Genetic testing for presence of *UGT1A1*1,*28, *36, *37* variants	Irinotecan (DPWG [3,4], French Group of Clinical Onco-pharmacology (GPCO-Unicancer) & National Pharmacogenetics Network (RNPGx) [30])	Dosing adjustment to prevent serious ADR (hematological/gastrointestinal toxicity)
*UGT1A6*	Genetic testing for presence of *UGT1A6*4* (rs17863783) variant	Daunorubicin, doxorubicin (CPNDS [28])	Pediatric patients: Dosing adjustment to prevent serious ADR (cardiotoxicity)
*VKORC1*	Genetic testing for homozygous *VKORC1* rs9934438 status	Acenocoumarol, phenprocoumon (DPWG [3,4])	Increase attention/monitoring dose
	Genetic testing for presence of *VKORC1* rs9923231 variant	Warfarin (CPIC [9], CPNDS [11], DPWG [3])	Dosing adjustment for optimal efficacy (avoiding excessive bleeding/clotting)

^1^ Clinical genetic testing options can be found in the Genetic Testing Registry: https://www.ncbi.nlm.nih.gov/gtr/. ^2^ While data are emerging in support of other pharmacogenomic variants (particularly in non-White populations), they are not included in the table unless specified in an existing guideline, or in a drug label. ^3^ Clinical guideline information can be found at https://www.pharmgkb.org/guidelineAnnotations. Annotations in this column specify guidelines associated with each drug or drug class, along with citation(s). For any that do not have an associated guideline, the drug labelling agency that has included the requirement for genetic testing in that drug’s label is listed. Notes: Gene–drug pairs are not included if associated guidelines conclude with no actionable recommendations. ADR = adverse drug reaction; DPWG = Dutch Pharmacogenetic Working Group; CPIC = Clinical Pharmacogenetic Implementation Consortium; CPNDS = Canadian Pharmacogenomics Network for Drug Safety; SCAR = severe cutaneous adverse reactions (including drug hypersensitivity syndrome, Stevens–Johnson syndrome, toxic epidermal necrolysis, drug reaction with eosinophilia and systemic symptoms, and maculopapular exanthema); SDM = shared decision making; UM = ultrarapid metabolizer.

**Table 3 jpm-09-00040-t003:** Advantages and disadvantages of various clinical genetic testing approaches for pharmacogenes in germline DNA—in general, and considerations for complex genes such as *CYP2D6.*

Approach	General Considerations		Additional Considerations for Complex Genes (e.g., *CYP2D6*) ^1^
	Advantages	Disadvantages	
Real-time PCR (RT-PCR) with Taqman probes	Efficient: Amplification and interrogation occur in one step Identifies only variants of known significance Identifies only variants in target genes	Cannot discover novel variants Taqman assay primers and probes are proprietary and informational detail about them is thus not accessible—complicating result interpretation in rare cases	Include TaqMan assays for copy number variation (CNV) *CYP2D6:* TaqMan *CYP2D6* gene copy number assay(s)—Applied Biosystems, Foster City, California, USA
Restriction Fragment Length Polymorphism (RFLP) analysis	Low cost—good for population health/clinical applications	Lower sensitivity—will detect 90–95% of variants, versus 99% [76] Slow and cumbersome The technology for RFLP testing has remained largely unchanged for the past two decades	Long-range PCR (XL-PCR), a challenging technique, may be required for pre-processing samples
Microarray (e.g., the well-established Amplichip CYP450 (Roche) test which is based on Affymetrix array technology	Relatively low cost Efficient—high throughput analysis Good for clinical laboratory settings Able to detect both SNPs and CNVs	Low discovery power—restricted by variants included in the assay Lower sensitivity—will detect 90–98% of variants, rather than 99% [77]	Higher sample quality/DNA integrity required for deletion/duplication analysis Microarray approaches will not detect hybrid genes unless specific primers are used to amplify hybrid gene(s) and if hybrid-gene-specific probes are included in the design of the microarray [78]
PCR + Sanger sequencing	Sanger sequencing is gold standard for verification of variants Can be more cost effective for small number of samples	Slower and relatively more cumbersome More expensive—particularly for large sample sizes	XL-PCR, a challenging technique, may be required for pre-processing samples
Multiplex PCR + Library preparation + Next Generation Sequencing			
*For gene panel*	Better discovery power versus RT-PCR Identifies only variants in target genes	Can miss discovery of novel variants, but better discovery power than RT-PCR	
*For exome sequencing (ES)/genome sequencing (GS)*	High discovery power	Identifies variants of unknown significance (VUS) Identifies secondary/incidental findings in genes unrelated to pharmacogenetics Less cost-effective and more time-consuming relative to sequencing panel targeted to pharmacogenes Less robust for the purposes of interrogating particular pharmacogenes, unless NGS libraries have been enriched for this purpose ES would likely have difficulty with hybrid genes and cannot identify variants outside—or not adjacent to—the exome	Concordance of *CYP2D6* results between ES and gene panel sequencing varies according to analysis parameters (e.g., >99% concordance with a truth-sensitivity threshold set at <99%, but <90% with a truth-sensitivity threshold set at <99.9%) [79] Concordance of *CYP2D6* results between GS and gene panel sequencing is lower compared to other genes (e.g., 90% rather than >97% [80]) due to lower coverage depth or variant calling difficulties for GS data
*Technology: Amplicon sequencing*	Requires smaller amounts of DNA Can be less cost effective for small number of samples	Limited discovery potential	Alignment of multiple short reads in the context of highly repetitive genes (such as *CYP2D6*) can result in higher error rate, with increased frequency of false negative/positive results
*Technology: Single-molecule real-time (SMRT) sequencing assay*	Good performance on identifying splicing isoforms	Expensive and lower accuracy compared to short-read sequencing	Uses long reads of the whole gene (e.g., *CYP2D6* [81]) and incorporates targeted sequencing of duplicated copies as necessary. This technique avoids a pitfall of many NGS platforms for complex genes—the misattribution of short reads to or from pseudogenes—however, it is more expensive and less accurate than other NGS approaches
*Technology: Nanopore sequencing*	Low capital cost Easy to integrate into clinical setting—palm-size portable equipment Fast turnaround time for results Can be more cost effective for small number of samples	Less efficient (lower throughput capacity) While nanopore sequencing has improved in accuracy over the past several years, it is not clear if it is sufficient for SNPs	Long-read nanopore sequencing for complex genes available (e.g., *CYP2D6* [82])

^1^ Given the complexity of the *CYP2D6* gene, including the presence of pseudogene homologous regions (*CYP2D7, CYP2D8*), hybrid genes, and gene duplications/deletions—all of which complicate the design of specific probes or primers—most studies recommend to start with XL-PCR for amplification of the whole locus in long segments [83,84]. For sequencing, amplification product(s) can then be fragmented or tagmented. Tagmentation is a rapid, easy-to-perform approach that combines the steps of fragmentation and adding adaptors to the ends of each fragment [85].

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
