# Peer review of "Pharmacogenomic Testing: Clinical Evidence and Implementation Challenges"

_jpm, 2019, doi:10.3390/jpm9030040_

Round 1

Reviewer 1 Report

The manuscript's revision addressed some of the significant issues that have been flagged by reviewers, but still remains quite biased towards PharmGKB as the primary source of reference for clinicians without proper explanation why this resource is more reliable and trustworthy than alternatives such as CPIC or the FDA's list of drugs with pharmacogenomic labeling.  This paper has very strong but somewhat conflicting statements such as on page 9 lines 146-153:

"There are also limitations to the information available on th PharmGKB website with respect to supporting the interpretation of pharmacogenetic test results –  discussed further under ‘Barrier 2’. The PharmGKB website is thus most useful to individual clinicians as an initial screening tool to identify whether pharmacogenomic testing may be worth considering for a given drug – by checking to see if there is a pharmacogenomic guideline with clinically actionable recommendations for that drug. Our strongest recommendation on an individual  level is for a physician to first consult PharmGKB and then consult with a pharmacogenomics specialist before ordering a pharmacogenomic test. "

Authors fail to acknowledge that some statistical associations listed on PharmGKB are not yet clinically actionable.   As an advocate for pharmacogenetic testing, I could encourage to apply much more cautious, and avoid such strong statements that are not yet supported by clinical guidelines - especially that only ~30% of PharmGKB's drug gene-pairs have independently reviewed clinical guidelines. PharmGKB is sill not a peer reviewed reference, and does not support meta analyses of frequently conflicting studies collated in the database.

Another limitation of PharmGKB is marker-specific "gene-drug" pairing rather than the diplotype-specific dosing recommendations as adopted by CPIC, is completely ignored by the authors.

Minor comments:

Table 1: suggest removing the first column in order to reduce complexity and content duplication - voriconasole is listed at list three times under different Clinical specialty areas

Reviewer 2 Report

Your review is well organized, and additional emphasis detailing the challenges of analyzing CYP2D6 within pharmacogenomics is a strength of the manuscript. Very timely and measured contribution. Some minor comments follow.

On page 6 within the box, section 2. Challenges to integration bullets, suggestion to drop ‘s’ in Lack of supports

Line 127: Suggest adding patient for individual patient level (otherwise it could be misinterpreted as provider, unless that is who “individual” is referring

In the section starting line 161 regarding community level, acknowledgment of commercial pharmacogenomics clinical decision support solutions existing to operationalize the clinical decision protocols and more quickly synthesize the available scientific evidence for actionability is missing. Consider including a point that provides a nod to such (e.g., Translational Software, and other such commercial entities)

Consider inclusion of Pharmacogenetics Standards Model in citations regarding section referencing guideline and protocol resources (http://www.nationalacademies.org/hmd/~/media/Files/Agendas/Activity%20Files/Research/GenomicBasedResearch/Action%20Collaboratives/DIGITizE/Pharmacogenetics%20Standards%20Model.pdf?la=en)

Line 194: Treat data as plural. Data are rather than data is (concede to editorial standards)

Line 221: Unsure whether PCR needs to be spelled out at first use here

Line 222: Recommendation to follow Jarvik and Evans terminology convention (PMID: 27657676) since the text here is describing tests used within the clinical context. Therefore, suggest dropping whole before whole genome and whole exome

Line 228: italicize gene name, CYP2D6

For Table 3, maintain formatting consistency with periods in the cells for the points included or drop all periods; add a comma after e.g. in nanopore sequencing row

Line 244: editorial preference for whether to not italicize gene when used as an adjective or modifier

Overcoming barrier 2 is nicely characterized.

Line 261 can you provide a citation for ClinVar; remove contraction and spell out

Line 389: suggest further specification for individuals; individuals who underwent genetic testing

Line 462: suggest word change for great to useful, as great does not provide an anchor to the magnitude in which the claim asserts

Round 2

Reviewer 1 Report

Authors addressed the major issues flagged in the previous review.

No further comments

This manuscript is a resubmission of an earlier submission. The following is a list of the peer review reports and author responses from that submission.

Round 1

Reviewer 1 Report

In this review, Hippman and Nislow discuss barriers to clinical implementation of pharmacogenomic testing. The first barrier involves deciding whether or not to order a test, and the second barrier involves logistical challenges of implementation.  However, many additional barriers have been outlined by others, and this review seems a little naïve, and perhaps misleading to potential readers who are unfamiliar with the field. Many of the conclusions drawn (examples discussed below) contrast work of other experts in the field, but the authors provide little to no evidence to support their conclusions.  Further, the authors state that they will discuss implementation in community settings, but the content of the manuscript is not well-focused and largely irrelevant to community-level pharmacogenomics implementation.

Major issues:

·         The authors reduce the many issues with implementation of pharmacogenomics into two barriers. Many in the field (including myself) would disagree with this assertion. Important additional barriers  In fact, CPIC was created for the express purpose of addressing the barrier of what to do with genetic information once it is available.  I would refer the authors to the following literature:

o   PMID: 23588301

o   PMID: 28090649

o   PMID: 27214750

o   PMID: 28619604

·         The authors seem to conflate being a PharmGKB “Very Important Pharmacogene” with having guidelines for clinical implementation.  PharmGKB VIPs do not all have guidelines – likely because there is not sufficient evidence to guide therapy for any particular drug.

·         Table 1: Only a subset of the listed drug-gene pairs have published guidelines from groups like CPIC or DPWG. There’s no indication of which gene-drug pairs are well-supported by the literature and which are not. Further, there are no citations or links to guidelines or evidence supporting the information provided. Lastly, the table contains at least one error – clopidogrel is listed as an anti-coagulant.

·         Importantly, CPIC explicitly states that its guidelines are not meant to answer the question of whether or not to order a pgx test, but instead provide information on how to use the data if available. This should be clarified in the discussion of CPIC guidelines in the “Should a pharmacogenomic test be ordered?” section.

·         Table 2: Similar to table 1, this table includes information beyond the stated scope of the review, and provides little useful information for potential community implementers. Perhaps a list of genes that are included in CPIC or DPWG guidelines would be more applicable.

·         Lines 93-95:  It is not clear who has decided that the drugs listed in Table 2 have recommendations appropriate for use in clinical care. Many experts in the field would argue that not all PharmGKB VIPs have sufficient evidence or recommendations to use clinically. There are certainly not clinical guidelines available for each of them as suggested.

·         Overcoming barrier 1: No new information is provided. After discussing problems with published guidelines, the reader is urged to reference published guidelines.

·         Line 147-166:  Suggesting authors of CPNDS recommendations of basing their recommendation on prospects of financial gain without evidence could provide a problem for both the authors and journal.

·         Table 3: Again, it is not clear who the intended audience is. For example, one of the disadvantages given is that some technologies cannot be used to discover novel variants; this is generally not something community practitioners are concerned with.

·         Discussion on next generation sequencing: It is not made clear how establishing a next-generation sequencing pipeline is a barrier to community implementation. On the subject of variants of unknown significance, the authors state that targeted sequencing is logistically and ethically contentious but do not provide a citation. They do not address the difference between finding a variant of unknown significance in a pharmacogene versus a gene associated with disease risk.  Along those lines, it is mentioned that sequencing is a cost effective method for genotyping for pharmacogenomics variants, yet no evidence is provided to support this assertion. 

Minor Issues:

·         Line 64: When stating that 509 drugs have pgx information in the label, the source should be cited.

Reviewer 2 Report

This is an excellent review of the issues involved with PGx implementation. My only suggestion is in the Overcoming Barrier 1 section, I do think some consideration should be given to the benefit of pre-test counseling and utilizing a shared decision-making model commonly practiced by genetic counselors in other specialties (e.g. oncology). It is my opinion that pre-test counseling in PGx is vitally important, and we should be encouraging our genetic counseling colleagues to give consideration to engaging in PGx. I also think it's important for other HCP to recognize that genetic counselors have the skills to discuss these issues, and employing genetic counselors in the process will ultimately improve patient care and increase efficiencies for the provider. Obviously, this is not the scope of this paper, and the authors do make a point about the value in a later paragraph, but I think pre-test counseling is a slightly different argument than supporting HCP when they receive an unexpected result. Also resources for finding genetic counselors (e.g. NSGC or CAGC) could be added to the executive summary.

Reviewer 3 Report

This manuscript contributes to the recent body of publications aimed to educate healthcare providers on clinical utility of PGx testing, and underlying genetic testing technologies. Authors provided excellent overview of pros and cons of various genetic testing platforms used for PGx testing, arguing that NGS (or better whole genome sequencing) is the ultimate solution for clinically relevant PGx testing and reporting. Although exome and whole genome sequencing becoming more cost effective, such testing will include significant number of genes that have substantial disease risk liability that requires expert patient counselling - for example the PharmGKB VIPs includes BRCA1 and BRCA2, thus patient with positive results may have substantially increased risk of cancer. Authors should address what types of healthcare providers would be responsible for communicating test results to patients. It is unrealistic to expect primary care physicians to address the whole spectrum of disease and drug risk liabilities. Major issues: 1. Authors list two main barriers to clinical implementation of PGx testing, but largely ignored the issue of clinical acceptance of PGx recommendations by physicians. Recommend to address physician acceptance as Barrier #3, especially when authors already commented on some of these aspects. 2. The review is slanted towards genetic testing technologies, while addressing the key role of bioinformatics and consistency of functional annotations, authors refer to PharmGKB as the primary source of information for physicians. Such strong reference early in manuscript might be misleading for a non-expert user. Missing an explanation the on type of tissue used for PGx testing and reportable gene-drug pairs. Many of the genes listed in PharmGKB VIP table such as BRAF, ALK and others are mainly relevant when a tumour tissue biopsy is analyzed rather than normal somatic tissue. PGx test analyzing spit sample or whole blood DNA should not report on such gene-drug interactions. Secondly, vast majority of drug annotations in PharmGKB are just for specific SNPs and missing an integrated annotation for phased dilotypes. Using single SNPs for clinical decision making in many cases may lead to to inconsistent annotations and potentially incorrect clinical interpretation. For example, the ABCC1 gene has over dozen significantly associated SNPs for multiple medications, and different testing platform may report different SNPs, thus making it difficult to evaluate clinical implications of different tests. 3. Although authors mention the challenge of combining results from multiple genes, this point must be discussed earlier in the paper to better explain the limitations of the PharmGKB. Lack of validated frameworks to provide clinically relevant dosing guidelines for drugs that have statistically significant association with markers in different genes reported in PharmGKB, such as the antihypertensive medications, poses major challenge in using this source for direct support of clinical prescribing decisions. Authors should better explain difference between clinically actionable genetic associations vs statistically significant SNP associations. Given the issues highlighted above suggest to remove the reference to PharmGKB in the Conclusions section. Required edits: Table 1: FVL - change to appropriate gene name F5, and add F2 because both contribute to the risk ivacaftor - belongs to Respirology, not Gastroenterology voriconazole - appears too many times, Please pick one main clinical area Table 2: FVL - change to appropriate gene name F5 Table 3: for whole genome, clinical validation of variants and CNVs should be used. This point is addressed in the manuscript, and addition to table is needed.